# High-Fidelity Router Emulation Technologies Based on Multi-Scale Virtualization †

**He Song [1] , Xiaofeng Wang [1,2,*] , Mengdong Zhai [1] and Guangjie Zhang [1]**

[1]  School of Internet of Things Engineering, Jiangnan University, Wuxi 214122, China;
     6171910007@stu.jiangnan.edu.cn (H.S.); 6161914040@vip.jiangnan.edu.cn (M.Z.);
     6161914041@vip.jiangnan.edu.cn (G.Z.)

[2]  The Cyberspaces Security Research Center, Peng Cheng Laboratory, Shenzhen 518055, China

*   Correspondence: wangxf@jiangnan.edu.cn

†   This paper is an extended version of our paper published in the 2018 IEEE 7th International Conference on
    Cloud Networking (CloudNet), "Research on High-Fidelity Router Emulation Technologies Based on
    Cloud Platform".

**Abstract:** Virtualization has the advantages of strong scalability and high fidelity in host node emulation. It can effectively meet the requirements of network emulation, including large scale, high fidelity, and flexible construction. However, for router emulation, virtual routers built with virtualization and routing software use Linux Traffic Control to emulate bandwidth, delay, and packet loss rates, which results in serious distortions in congestion scenarios. Motivated by this deficiency, we propose a novel router emulation method that consists of virtualization plane, routing plane, and a traffic control method. We designed and implemented our traffic control module in multi-scale virtualization, including the kernel space of a KVM-based virtual router and the user space of a Docker-based virtual router. Experiments show not only that the proposed method achieves high-fidelity router emulation, but also that its performance is consistent with that of a physical router in congestion scenarios. These findings provide good support for network research into congestion scenarios on virtualization-based emulation platforms.

**Keywords:** cyberspace security; network emulation; router emulation; traffic control; virtualization

---

## 1. Introduction

The research of cyberspace security for large-scale networks is an essential direction to ensuring the order of the Internet. But limited by the number of physical devices, it is not realistic to use physical networks for large-scale cyberspace security technology verification and offensive-defensive drills [1,2]. Emulation technology based on virtualization has the advantages of high controllability and excellent scalability [3,4], so it has become a significant tool to reproduce complex or large-scale network topologies. At present, researchers design virtual routers by loading routing software into virtual machines (VM) [5,6], and using Linux Traffic Control (TC) to control the bandwidth on a VM's network interface card (NIC) to emulate the transmission bandwidth of the physical Ethernet (e.g., 100 Mbps or 1000 Mbps) [7,8].

In this context, it is critical that the constructed virtual router has the same effect as the physical hardware, which is also called high fidelity. However, when emulating a large-scale denial of service (DoS) (such as a distributed DoS (DDoS) [9] or a low-rate DoS (LDoS) [10]), we found that the performance of the virtual router mentioned above is completely different from that of a physical router. For example, Wang et al. [11] emulated an LDoS attack [12] on a BGP session running autonomous systems (AS). In 50 experiments, the BGP session on the physical router with 1 Gbps NIC bandwidth

was successfully interrupted 14 times, whereas the TC-defined virtual router with 1 Gbps bandwidth was never interrupted (and hence, the BGP session was never reset). The cause of such distortion comes from the token bucket scheme in TC. In congestion sensitive scenario, compared with the physical device, TC improves the probability of transmitting short packets, such as BGP keep-alive packets. This has a large impact on the emulation results, leads to the illusion that DDoS or LDoS attacks cannot interrupt BGP sessions, and also shows the low-fidelity of the TC-based virtual router.

In order to solve this problem, we propose a high-fidelity router emulation scheme that can effectively control the bandwidth and perform exactly the same as a physical router. Different from TC, we no longer use token bucket as the method of bandwidth control, but a delay waiting method to accurately calculate and control the transmission time of each waiting packet in the buffer queue, which is closer to the performance of physical router. Especially in congestion scenarios, our virtual router successfully emulates real LDoS and DDoS attack phenomena.

In addition, the emulation scheme above has been roughly introduced in our first version [11], and we have constructed a module and verified it based on the kernel of KVM. However, according to further research, for one thing, KVM has some limitations, such as large resource consumption and poor migration. For another, there are several different virtualization methods for router emulation, and the kernel programs of some lightweight virtual routers (such as Docker-based virtual router) cannot be modified because they share the same kernel space with the host. Therefore, we applied our emulation scheme in the user space of Docker based on *iptables* (a firewall software on Linux) and *libnetfilter_queue* (a user space network library). Then, we used two different virtualization methods to realize multi-scale virtualization.

Overall, the main contributions of our paper are as follows:

1. We propose a high-fidelity router emulation scheme that consists of virtualization plane, routing plane, and a traffic control method. We focus on the composition of a traffic control method, which uses the "drop from tail" algorithm as the buffer queue management, the first in, first out (FIFO) method as the buffer queue scheduling rule, and the delay waiting method as the bandwidth control module. The specially designed bandwidth control module can solve the distortion of virtual router.
2. We customized our traffic control module separately in the Network Protocol Stack in the Linux kernel of KVM and the user space of Docker, building a KVM-based virtual router and a Docker-based virtual router. Docker has many advantages, such as a light weight, small resource occupation, and convenient migration. It helped us extend the router emulation scheme to multi-scale virtualization, and greatly expands the scale of emulation topology.
3. To verify the fidelity and practicality of our KVM and Docker virtual routers, we integrated the two virtualization methods, Docker and KVM, built a complex inter-AS network topology with 3000 virtual routers and 5179 links, and successfully simulated LDoS attack behavior. This proves the effectiveness of our emulation method, and it is also meaningful for improving the emulation fidelity in congestion scenarios.

The remainder of this paper is organized as follows: In Section 2, we introduce the current router emulation scheme. In Section 3, we present our router emulation architecture and novel traffic control method, which improves the fidelity of the virtual router. Based on the proposed architecture, in Section 4, we implement our traffic control module in the kernel space and user space of the virtual router. Then, in Section 5, we evaluate our scheme on the basis of traffic control effectiveness and router emulation fidelity.

## 2. Related Work

Researchers have explored a number of different technologies and proposed a variety of solutions to construct a complete router emulation. Guo et al. [13] attempted to achieve router emulation by combining Click and NS2. However, their virtual router does not meet the fidelity requirements of

emulation experiments because NS2 is simply a discrete event simulator. Hou et al. [14] used Quagga as the routing engine and OpenFlow as the virtual switch to complete data forwarding. They also deeply customized the Floodlight (SDN controller) to complete their virtual router. They focused on the interconnections between the virtual router and the traditional network, but the interconnection performance between virtual routers was not sufficiently robust for network emulation experiments. Based on VegaNet, Zhang et al. [15] combined Xen semi-virtualization technology, Quagga, and Click to construct a virtual router. However, the use of VegaNet makes the router incompatible with mainstream virtual network systems, such as OpenFlow. This made it is difficult to migrate their virtual router to cloud platforms. Based on KVM and LXC virtualization respectively, Kamla et al. [16] used the routing software Quagga for implementation, which meets the routing and forwarding requirements of cloud platforms, and thus, is a commonly used router emulation method in the current virtual environment. But their route emulation method did not consider bandwidth control, which makes the traffic between virtual links uncontrollable and unable to emulate congestion scenarios. Our virtual router combines the work of [6,16], and adds traffic control on that basis.

In general, link traffic control is realized based on TC in virtualized network emulation environments [7,8]. TC is a flexible and powerful traffic control method built on the Network Protocol Stack of the Linux kernel to ensure the quality of service (QoS) of the Linux system. It includes a series of flow control strategies (e.g., FIFO, PFIFO_fast, TBF, HTB, and red [17]). PFIFO_fast is the default strategy adopted by Linux. PFIFO_fast allocates data packets to three FIFO queues based on their type of service (TOS), and sends as many packets as possible without any bandwidth control, while TBF and HTB are the most accurate traffic control strategies in Linux and can ensure that the bandwidth of a NIC remains below a certain value [18]. They are often used in TC for traffic control in network emulation. For example, based on the emulation scheme proposed by Kamla et al. [16], Liu et al. [19] and Mendoza et al. [20] emulated satellite routing. Furthermore, they successfully emulated link characteristics such as bandwidth, delay, and packet loss rate in satellite link emulation with TC-HTB. However, Wang et al. [11] (our first version) found that TC-HCB did not perform well in congestion scenarios, and it could not replicate the real situation of BGP-LDoS attacks on virtual platforms. They proposed a congestion control scheme for queuing delay emulation to solve this problem, but they did not elaborate on the details of the solution and verified it only on a KVM-based virtual router.

Based on multi-scale virtualization, this paper focuses on the fidelity of router emulation. We studied how to build a virtual router and designed a traffic control module based on KVM and Docker. We hope our modules are better able to support network emulation research.

## 3. Router Emulation Architecture

Figure 1 shows the architecture of our virtual router. Based on the physical server and operating system, we used Docker and KVM in the virtualization plane, Quagga in the routing plane, and designed our own traffic control module. The coverage of different virtualization planes forms multi-scale virtualization. Specifically, we provide detailed introductions in Sections 3.1 and 3.2.

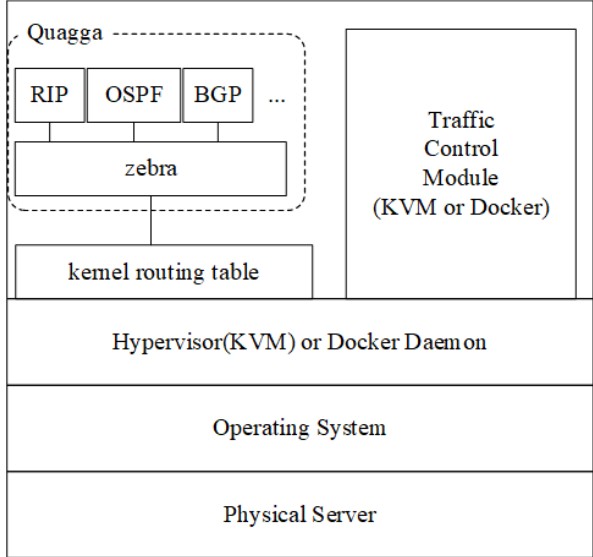

**Figure 1.** Virtual router architecture.

### 3.1. Virtual Router

As a result of its high-fidelity, virtualization techniques are widely used in router emulation. In this paper, the term "virtual router" includes both a Docker-based virtual router and a KVM-based virtual router.

(1) Virtualization plane: KVM is a virtualization module embedded in the Linux kernel [21]. A KVM-based VM has an independent kernel space; hence, we can freely customize the kernel files in the VM to satisfy the router emulation requirements. KVM-based VMs also boast a high degree of system isolation and an excellent emulation fidelity. However, KVM-based VMs require a relatively large amount of hardware resources, and it is difficult to construct a large-scale virtual node emulation.

Docker is an OS-level virtualization solution based on Linux Containers (LXC) [22]. The containers created by Docker are superior in several ways, such as their fast startup speed, low resource consumption, and large deployment scale. However, all containers share the same host kernel and portion of the run-time library; consequently, the router emulation module cannot be customized by modifying the kernel [23].

Therefore, in Figure 1, we show our use of KVM-based virtual routers to emulate pivotal routers and Docker-based virtual routers to expand the emulation scale.

(2) Routing plane: Quagga is a routing software suite that supports protocol emulations, including RIP, OSPF, and BGP, and has a strong affinity for Unix-like systems [24]. Quagga is mainly composed of a routing protocol controller, a kernel routing table manager, and a kernel routing table. The routing protocol controller is responsible for configuring the routing protocols, for discovering the routes, and for calculating the paths of those protocols. The kernel routing table manager is responsible for adding the routing protocol configurations to the system kernel routing table; subsequently, the kernel routing table builds a data forwarding table and performs data forwarding. Therefore, in Figure 1, we show our use of scalable and modular routing software to implement support for multiple routing protocols in router emulation.

### 3.2. Traffic Control

#### 3.2.1. Problem Statement

TC uses FIFO, drop from tail, and token bucket strategies for traffic control. Specifically, all the packets enter the buffer queue following a FIFO scheduling rule, and the drop from tail queue management policy drops packets when the buffer queue is full. At the same time, the system sends

tokens to the token bucket at a limited rate. When a packet is about to leave (to be sent) the buffer queue, it needs to consume the same number of tokens as its length. If the number of tokens in the bucket does not satisfy transmission of the packet, the packet will continue to wait for more tokens to be generated during its delayed transmission time. Finally, if the delayed transmission time has elapsed, the packet is dropped.

For example, we assume that the number of tokens generated per millisecond (ms) is 10, and that the delayed transmission time of each packet is 10 ms. If no token exists in the current bucket and the length of the packet to be sent is greater than 100, the packet will never obtain a sufficient number of tokens and will be dropped after 10 ms; conversely, if the length of the packet to be sent is less than 100, the packet can be successfully sent after waiting for some period of time. This case clearly demonstrates that if the data traffic remains excessively large, longer packets will be continuously dropped, and shorter packets will be successfully transmitted because it is easier for shorter packets to obtain enough tokens. This also increases the likelihood that subsequent packets will enter the buffer queue.

The purpose of BGP-oriented DDoS and LDoS attacks is to force keep-alive packets (with a length of 19) between BGP sessions to drop continuously, which will cause route flapping and a decline in network performance, and force the BGP session to reset. When using the TC strategy to control the emulation bandwidth of a link, a large data flow has little impact on the probability of dropping a keep-alive packet; thus, the BGP session never gets interrupted and reset, which causes the illusion that DDoS and LDoS attacks cannot interrupt a BGP session.

### 3.2.2. Traffic Control Method

We propose a high-fidelity traffic control method based on two aspects: queue scheduling and management, and bandwidth control. Queue scheduling refers to the process of arranging or rearranging packets in the queue to control their sending sequence [25,26]. Queue management is responsible for determining whether a received packet should be dropped [27]. In bandwidth control, for example, the token bucket algorithm [28], is used to control the transmission rate of packets below a fixed value. For the specific implementation of the two traffic control modules in Figure 1, see Section 4.

(1) Queue scheduling and management: Normally, the default queue scheduling method used by the physical router is FIFO, and the default queue management method is the drop from tail algorithm, which utilizes fewer resources to manage the packets. To ensure emulation fidelity, we also use FIFO and the drop from tail algorithm as the scheduling and management methods, respectively, for the virtual router's packet buffer queue. Figure 2 depicts the enqueuing and dequeuing processes of a packet in a buffer queue.

In detail, the sequence in which packets arrive is used as the sequence of the buffer queue. When a packet arrives, it will be inserted at the tail of the queue, and the packet at the head of the queue will be simultaneously sent to the network. When the buffer queue is full, newly arrived packets will be dropped until the queue has sufficient free space to receive the incoming packet.

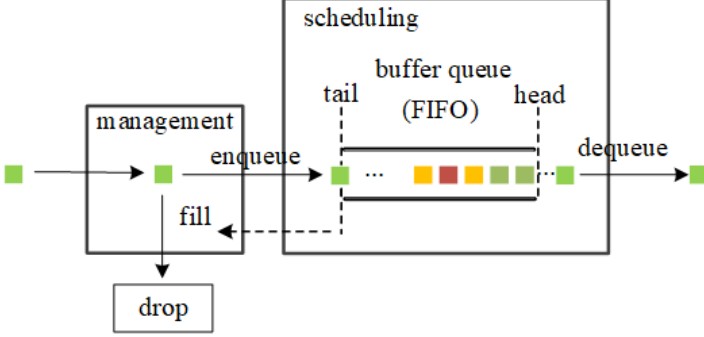

**Figure 2.** Queue scheduling and management.

(2) Bandwidth control: Bandwidth is usually defined as the maximum amount of data (measured in bps) that the NIC can send or receive in a given period of time. In general, we can limit only the packets sent by the NIC; we cannot limit the packets received. Therefore, the process of controlling bandwidth involves precisely controlling the total size of the packets that the NIC can send per unit of time.

Assuming that the bandwidth of the NIC is *B*, for a packet of length *L*, the theoretically required transmission time *T* can be calculated as follows:

$$T = L/B. \tag{1}$$

In a virtual environment, because the I/O operations of a VM are extremely fast, we assume that the virtual router NIC actually sends packets within a negligible transmission time. Therefore, we set a time delay *T* before each packet is sent and then deliver the packet to the NIC for packet transmission to achieve bandwidth control of a virtual router.

Algorithm 1 describes our bandwidth control method in detail. First, the algorithm calculates the packet length *L* of the head packet *packet_skb*. According to the preset bandwidth *B*, the theoretical transmission time *T* of the packet is calculated by Formula (1). Then, the algorithm creates a high-precision timer and delays the transmission by *T*. Finally, the NIC driver sends the packet to the physical layer for further transmission.

---

**Algorithm 1. Bandwidth control algorithm.**

---

**Input:**
    *packet_skb*, *B*;
    //*packet_skb* is the packet dequeued from the FIFO queue header; *B* is the bandwidth value that we need to emulate.
**Output:**
    *packet_skb*;
1:   $L \leftarrow$ Obtain the length of *packet_skb*;
2:   $T \leftarrow L/B$;
3:   $T_{start} \leftarrow$ Retrieve the current system time;
4:   $T_{end} \leftarrow T_{start}$;
5:   **while** $T_{end} - T_{start} < T$
6:      $T_{end} \leftarrow$ Retrieve the current system time;
7:   **end while**
8:   return *packet_skb*;
    //network interface driver sends the packet

---

## 4. Router Emulation Implementation

As described in Section 3.1, based on multi-scale virtualization and Quagga, we can easily implement virtual router. In this section, we focus on our traffic control modules based on KVM and Docker to achieve high-fidelity router emulation.

### 4.1. KVM-Based Traffic Control Module

The KVM-based Linux VM has its own independent kernel space. In the Linux kernel, TC provides traffic control policies through Qdisc (a network scheduler). Therefore, we implemented the traffic control method described in Section 3.2.2 in the TC module within the Network Protocol Stack of Linux 3.2.90, and we used Qdisc to associate our traffic control policy with the NIC to implement the bandwidth emulation on the virtual router.

As shown in Figure 3, our traffic control module includes three parts: queue management, queue scheduling, and bandwidth control.

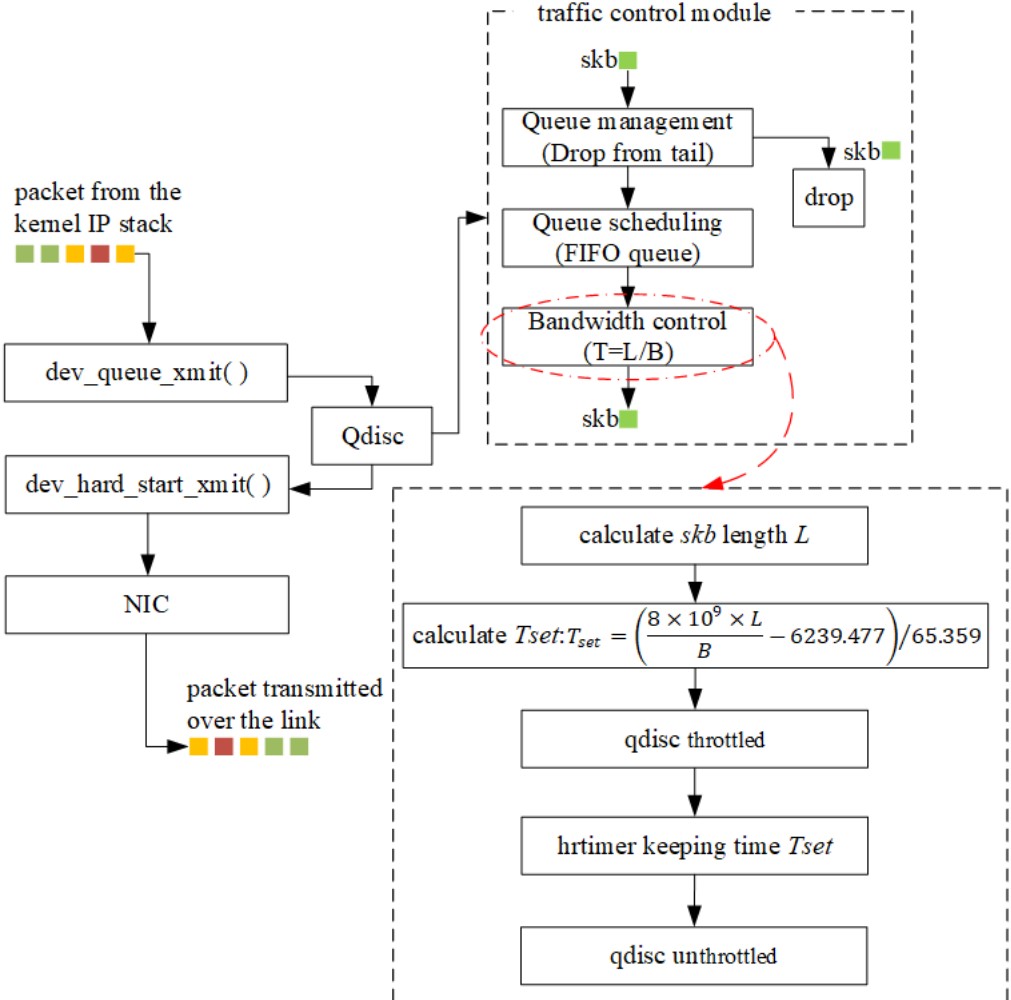

**Figure 3.** KVM-based traffic control module architecture.

1.　Queue management: This module is responsible for receiving the packets delivered by the Linux kernel IP Stack and determining whether each packet enters the queue scheduling module or is dropped according to the drop from tail algorithm.
2.　Queue scheduling: This module is designed to initialize the FIFO buffer queue and sort the packets entering the buffer queue according to the FIFO principle.
3.　Bandwidth control: This part is the most important module. It obtains each dequeued packet from the queue scheduling module and calculates its delay time $T$. By calling the *hrtimer* (a high-resolution timer) in the kernel to perform the delay operation, the bandwidth control module can emulate the packet transmission situation under the available bandwidth $B$.

However, in an actual emulation, most program operations suffer from certain systematic errors. Since the delay time of a data packet is usually in the order of nanoseconds, a slight systematic error between the set delay time $T_{set}$ and the actual delay time $T_{real}$ can lead to a large bandwidth control error. To solve this problem, we obtain multiple data pairs of $T_{set}$ and $T_{real}$ through experiments and fit these discrete data to capture the linear relationship between $T_{set}$ and $T_{real}$. This linear relationship is shown in Figure 4. The regression formula is as follows:

$$T_{real} = 65.359 \times T_{set} + 6239.477. \tag{2}$$

Under a preset NIC bandwidth *B*, for a packet of length *L*, we deduce $T_{set}$ according to Formula (2):

$$T_{set} = (\frac{8 \times 10^9 \times L}{B} - 6239.477)/65.359, \tag{3}$$

where $T_{set}$ is measured in *ns*, *L* is the length of the packet in *bytes*, and *B* is measured in *bit/s*. This dynamic adjustment of $T_{set}$ offsets the systematic error and stabilizes the control of the bandwidth *B*, thereby achieving high-fidelity bandwidth control of the KVM-based virtual router. Note that the constant terms in Equations (2) and (3) are obtained in our experimental environment. Although we have tried many different hardware configurations and obtained very similar formulas, we still can not guarantee that the constants can be applied in all environments. Researchers should reconstruct the appropriate equations in their experimental environment.

After associating *Qdisc* with our traffic control module in the KVM-based virtual router, the packet process in the Linux kernel is as follows:

Step 1: *dev_queue_xmit()* (a function in *net/core/dev.c*) is used to obtain the packet *skb* waiting to be sent in the Linux IP Stack.

Step 2: The function *dev_queue_xmit()* delivers *skb* to *Qdisc* to complete traffic management.

Step 3: *skb* enters our traffic control module through *Qdisc*.

Step 4: Our queue management module first determines whether the current buffer queue is full. When the current queue buffer is full, the module directly returns a command that *skb* is to be dropped; otherwise, *skb* is delivered to the queue scheduling module.

Step 5: The queue scheduling module inserts *skb* into the tail of the queue and updates the backlog size of the queue.

Step 6: When possible, the queue scheduling module takes the packet *skb* from the head of the queue and delivers it to the bandwidth control module.

Step 7: The bandwidth control module calculates the packet length *L* and calculates $T_{set}$ according to Formula (3).

Step 8: The *hrtimer* is used to implement a delay time $T_{set}$.

Step 9: The packet *skb* is returned to *dev_hard_start_xmit()* through *Qdisc*.

Step 10: *dev_hard_start_xmit()* delivers *skb* to the NIC driver to send the packet.

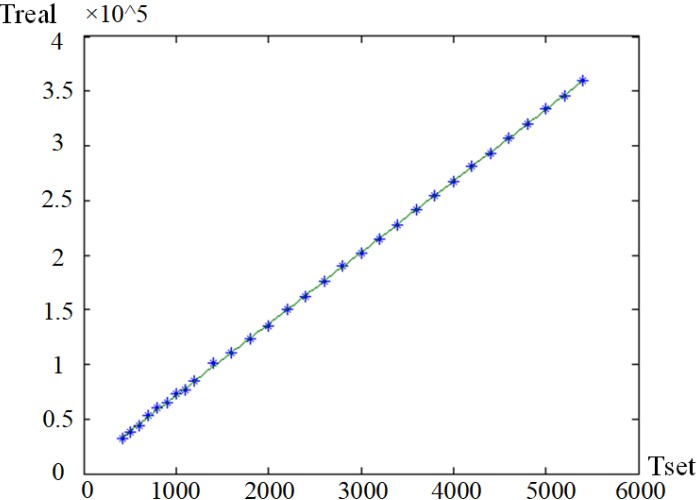

**Figure 4.** KVM-based $T_{real}$ and $T_{set}$ fitting results.

### 4.2. Docker-Based Traffic Control Module

Docker is an OS-level virtualization technology. The Docker-based virtual router shares the same host kernel with other containers; therefore, we cannot control the transmission of packets through

the TC module in the Linux IP Stack. Instead, we need to design a high-compatibility traffic control module based on user space.

The Netfilter framework provides a variety of solutions, such as *iptables*, *libnfnetlink*, and *libnetfilter_queue*, for controlling network traffic in user space. Some of the functions in Netfilter enable us to control the kernel to perform specified operations and achieve accurate traffic control. In detail, all the data packets sent by a virtual router will pass the *POSTROUTING* chain of the *mangle* table in the *iptables* before they leave the NIC. The Netlink socket implements inter-process communication (IPC) between the kernel and user space, thereby providing interfaces so that users can inter-operate with the kernel. On this basis, we can read the packets in the *POSTROUTING* chain, execute our traffic control strategy, and return the packet judgments of the to the kernel to achieve control of each packet.

As shown in Figure 5, we build a Docker-based traffic control module. First, we establish a queue (labeled as queue-num) for a NIC on a virtual router and execute the *NFQUEUE* rule to judge all the packets. In this scheme, each packet is allocated to the queue corresponding to its target NIC. Then, we create and initialize the Netlink socket to obtain packet information from the queue and provide our module's decision; that is, whether the packet should be accepted or dropped to the Netfilter framework in the kernel through IPC.

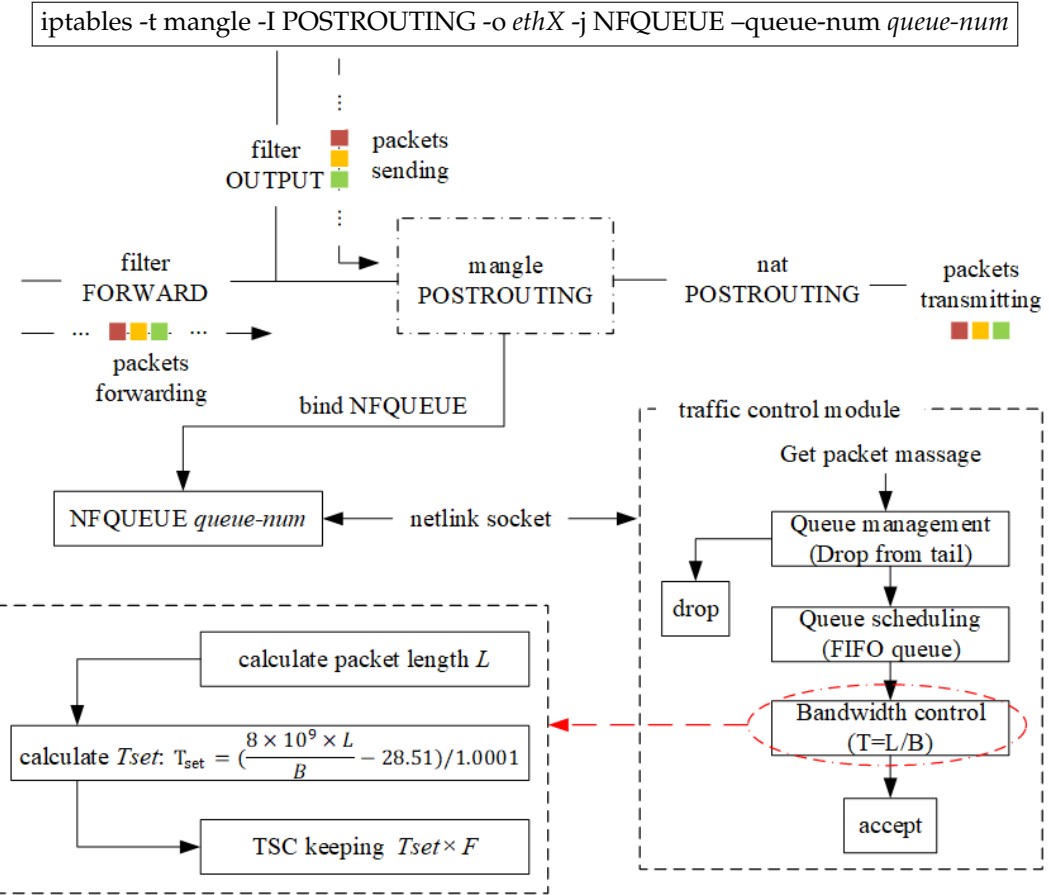

**Figure 5.** Docker-based traffic control module architecture.

A high-precision timer is required to implement a time delay for each packet. However, the *hrtimer* in kernel space cannot be accessed from user space. After many experiments, we chose the time stamp counter (TSC) to achieve precise timing by counting the number of clock cycles. Specifically, the value of the TSC is incremented by one at every clock cycle. We assume that the CPU frequency of the Docker host is *FHz*, (the precise value can be obtained by checking the CPU information); consequently, here, the clock frequency is $1/F$. To execute the time delay, we first obtain the current value $t_{start}$ of the TSC

using *RDTSC* (an assembly instruction), and then continually acquire subsequent TSC values $t_{end}$ until $t_{end} - t_{start} \geq T \times F$. However, a systematic error still exists between the set delay time $T_{set}$ and the actual delay time $T_{real}$ realized by the TSC. Through multiple sets of experiments, we obtained a linear relationship between $T_{set}$ and $T_{real}$. This relationship is shown in Figure 6, and the corresponding formula is:

$$T_{real} = 1.0001 \times T_{set} + 28.51. \tag{4}$$

Similar to the processes used to obtain Equations (2) and (3), the set delay time $T_{set}$ can be expressed as follows:

$$T_{set} = (\frac{8 \times 10^9 \times L}{B} - 28.51)/1.0001, \tag{5}$$

where $T_{set}$ is measured in ns. Eliminating systematic errors in this fashion stabilizes the bandwidth control of the Docker-based virtual router. Note that the constant terms in Equations (4) and (5) are obtained in our experimental environment.

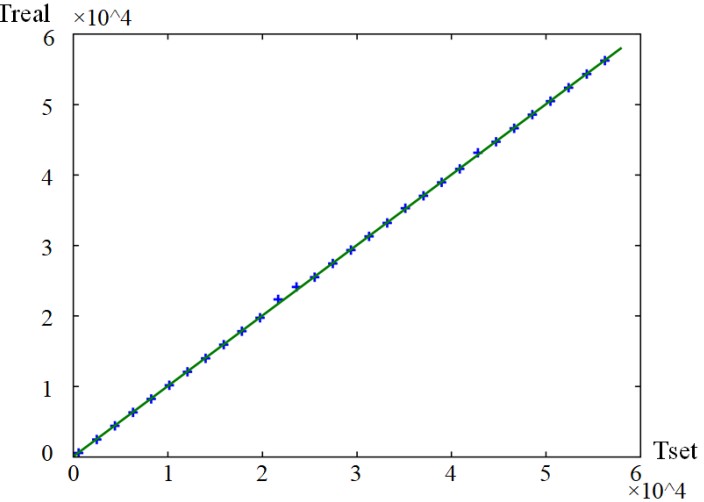

**Figure 6.** Docker-based $T_{real}$ and $T_{set}$ fitting results.

## 5. Experiment and Evaluation

To verify the effectiveness of our traffic control strategy and the fidelity of router emulation in offensive-defensive scenarios, we conducted the following four experiments:

1.  We evaluated the error between the set bandwidth and the actual bandwidth of the virtual router to demonstrate the effectiveness of our traffic control method.
2.  We compared the packet loss rate of the virtual router with that of a Cisco physical router to demonstrate the fidelity of our traffic control method.
3.  We compared the results of emulated BGP-DDoS and BGP-LDoS attacks in a physical network topology, a TC-based virtual topology, and a virtual topology based on our method to further validate the fidelity of our router emulation.
4.  Based on the OpenStack platform, we constructed a large-scale LDoS emulation scenario including both the KVM-based virtual router and the Docker-based virtual router to demonstrate the importance and value of our high-fidelity router emulation research in the network emulation domain.

### 5.1. Experimental Environment

We executed our KVM-based virtual router and Docker-based router on OpenStack Mitaka; the simulation consisted of one controller node, one network node, and several compute nodes. Each compute node was virtualized using Docker and KVM. The physical server hardware consisted

of a typical Dell PowerEdge R730 server with two Intel(R) Xeon(R) E5-4620 v4 processors and 64 GB of RAM. All the physical servers were interconnected via a 10 Gigabit switch and were running a Linux CentOS 7 operating system. The physical router type used in the experiment was a Cisco 4400.

## 5.2. Bandwidth Evaluation

We forced the virtual router to forward a large number of User Datagram Protocol (UDP) packets with random length and used the KVM-based and Docker-based traffic control modules to control the bandwidth. We set the traffic of the UDP packets to 3 Gbps to emulate a congestion scenario and tested several emulation bandwidths. The actual bandwidth performances and relative errors are shown in Table 1.

According to Table 1, under the different emulation bandwidths, when we use the KVM-based traffic control module to emulate the bandwidth, the maximum error between the set bandwidth and the actual bandwidth is 6.4%; the minimum error is 3.2%; and the average error is 4.95%. When we use the Docker-based traffic control module, the maximum, minimum, and average errors are 5.6%, 4.0%, and 4.76%, respectively. Our traffic control modules constructed on KVM and Docker boast a high bandwidth control accuracy and provide a high-fidelity emulation solution for virtual routers.

**Table 1.** Bandwidth performance.

|  | KVM-Based | | Docker-Based | |
| --- | --- | --- | --- | --- |
| Set Bandwidth | Bandwidth | Error | Bandwidth | Error |
| 100 Mbps | 96 Mbps | 4.0% | 96 Mbps | 4.0% |
| 300 Mbps | 284 Mbps | 5.3% | 283 Mbps | 5.6% |
| 500 Mbps | 473 Mbps | 5.4% | 475 Mbps | 5.0% |
| 700 Mbps | 655 Mbps | 6.4% | 661 Mbps | 5.5% |
| 900 Mbps | 851 Mbps | 5.4% | 859 Mbps | 4.5% |
| 1 Gbps | 968 Mbps | 3.2% | 960 Mbps | 4.0% |

## 5.3. Loss Rate Evaluation

We further evaluated the fidelity of the traffic control module integrated into our virtual router to investigate the packet loss rate. We separately employed a physical router, the KVM virtual router, and the Docker virtual router to establish three experimental scenarios. For the physical router, we chose a Cisco router with an NIC bandwidth of 1 Gbps to build a test link. For the KVM and Docker virtual routers, we fixed the bandwidth of the test link to 1 Gbps using our traffic control module. We tested the packet loss rate of 100 ping packets after multiple sets of UDP traffic with different magnitudes (500 Mbps, 1 Gbps, etc.) passed through the test link. The experimental results are shown in Table 2.

**Table 2.** Packet loss rate performance.

| UDP Flow | Cisco Router | KVM-Based | Docker-Based |
| --- | --- | --- | --- |
| 0 bps | 0% | 0% | 0% |
| 0.5 Gbps | 0% | 0% | 0% |
| 1 Gbps | 22% | 21% | 20% |
| 1.5 Gbps | 53% | 50% | 54% |
| 2 Gbps | 71% | 70% | 69% |
| 2.5 Gbps | 80% | 80% | 79% |
| 3 Gbps | 82% | 83% | 81% |

Obviously, in all three scenarios, when the UDP traffic does not fully occupy the link bandwidth, no packet loss occurs. However, once the UDP traffic reaches or exceeds the link bandwidth, packet loss is observed. Under the seven traffic magnitude settings, our method always exhibits a performance

close to that of the physical router. Taking the packet loss rate of the physical router as the standard, the experimental errors of the KVM and Docker traffic control modules are both approximately 2%. This means that our modules also have a good fidelity with regard to the packet loss rate.

### 5.4. Evaluation by Emulated DDoS and LDoS Attacks

To determine whether the proposed traffic control method could solve the distortion problem of TC in a congestion scenario, we constructed a small-scale experimental topology for emulating DDoS and LDoS attacks. As shown in Figure 7, Host-1 to Host-10 were the attack nodes (the VMs responsible for the attacks); Host-11, Host-12 and Host-13 were the target nodes (the VMs being attacked); and Router-1 and Router-2 were the virtual routers belonging to two different ASes. BGP was configured between the two routers so that they could learn the routing information and forward packets. The interval times of keep-alive and hold time were respectively set as 60 s and 180 s. Ideally, if we perform a DDoS or LDoS attack on the link between these two routers, link congestion will form, and as a result, the keep-alive packets and retransmitted packets in BGP will be continuously lost, interrupting the BGP session and forcing it to reset, causing link instability. In network security, this kind of attack will not directly affect the confidentiality, integrity, and authenticity of information, but it will destroy the availability [12,29].

We constructed four experimental scenarios for detailed comparison, including one physical scenario and three virtual scenarios. For physical scenario, we used real commercial PCs as the host nodes and Cisco routers (whose Ethernet interfaces are 1 Gbps) as the router nodes. For virtual scenarios, we applied KVM and Docker as virtual routers respectively, and used our traffic control module for 1 Gbps bandwidth emulation (KVM-our and Docker-our). In particular, we added a comparison with native KVM virtual router with Linux TC(KVM-TC).

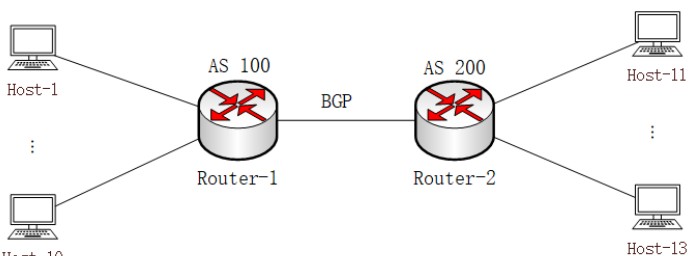

**Figure 7.** Topology of the distributed denial of service (DDoS) and low rate DoS (LDoS) experimental scenarios.

### 5.4.1. DDoS Emulation

Firstly, according to the fact that the traffic in BGP control plane and in data plane of the Internet use the same physical medium, we emulated three BGP-DDoS attacks with attack flows of 2 Gbps, 2.5 Gbps, and 3 Gbps. Specifically, Host-1 to Host-10 were set as bots and implanted with UDP traffic generators; then, they randomly sent a UDP data stream to Host-11, Host-12, and Host-13. The UDP data stream of a single bot was evenly distributed according to the set attack flows, and the duration of the attack was 10 min. Under each attack flow, we conducted 50 experiments and counted the number of BGP sessions dropped (the number of successful attacks). The experimental results are shown in Table 3.

**Table 3.** DDoS and LDoS emulation results.

| | Attack Flow | BGP Session Reset Time | | | |
|---|---|---|---|---|---|
| | | Cisco Router | KVM-TC | KVM-Our | Docker-Our |
| DDoS | 2 Gbps | 20 | 0 | 22 | 20 |
| | 2.5 Gbps | 39 | 0 | 38 | 38 |
| | 3 Gbps | 43 | 0 | 41 | 42 |
| LDoS | 2 Gbps | 5 | 0 | 5 | 4 |
| | 2.5 Gbps | 8 | 0 | 9 | 8 |
| | 3 Gbps | 14 | 0 | 15 | 16 |

In the physical scenario, 20, 39, and 43 BGP sessions are dropped between the two routers. In other words, as the DDoS attack flow increases, the success rate of the attack increases. At the same time, the two virtual scenarios constructed with our traffic control modules, KVM-our and Docker-our, displayed performances very close to that of the physical scenario. Under different attack flows, 22, 38, and 41, BGP sessions are dropped in the KVM-our virtual router, while 20, 38, and 42 BGP sessions are dropped in the Docker-our virtual router. The average numbers of errors between these two virtual routers and the physical router are both only one dropped session. However, in the KVM-TC virtual router constructed by the TC method, no BGP sessions are dropped, resulting in the illusion that DDoS attacks cannot successfully block the target link.

5.4.2. LDoS Emulation

Next, because BGP is an application layer protocol running on TCP, the TCP-targeted LDoS attack can effect BGP session. We emulated an BGP-LDoS attack with square wave flows of 2 Gbps, 2.5 Gbps, and 3 Gbps. In the experiment, the TCP connections uses uniformly recommended minRTO ($minRTO = 1$ s). For each bot, we created a periodic square-wave of UDP data stream with peak $R$, burst length $L$, and period $T$. $R$ was set to the average of the total square wave flows, $L$ was set to 0.6 s, $T$ was set to 1 s, and the attack duration was 30 min. As with the DDoS emulation, we performed 50 experiments in each of the four experimental scenarios. The results are shown in Table 3.

As the benchmark, 5, 8, and 14 BGP sessions are dropped in the physical router under the three different attack flows. The numbers of dropped BGP sessions in the KVM-our and Docker-our scenarios are basically consistent with those in the physical scenario, and the average error is approximately one dropped session, reflecting the ability of our virtual routers to emulate an attack with a high fidelity. In contrast, the BGP sessions still fail to break and reset under the KVM-TC scenario.

In summary, in the emulated DDoS and LDoS attacks, we reveal that our virtual routers essentially perform the same as the physical router. However, due to the defect caused by the token bucket in TC, when the link is congested, its packet loss performance is considerably different from that of the physical router; thus, its emulation fidelity is low. When using the traffic control method proposed in this paper to emulate the bandwidth of the virtual router, the module is not inclined to send short-length packets and discard long-length packets; hence, the designed traffic control module is highly consistent with the physical router. Furthermore, the errors in the DDoS and LDoS attack phenomena relative to the physical scenario are small, and the emulation fidelity is high.

*5.5. Evaluation of a Large-Scale LDoS Emulation*

In the previous section, we verified that DoS attacks, such as DDoS and LDoS attacks, can effectively cause BGP sessions between virtual routers to drop and reset. However, real-world network topology is usually highly complex. The purpose of a DoS attack is to cause the target link to drop and reset with a high frequency. In detail, once the BGP session of a target link is dropped, the routers at both ends of the link will recalculate the optimal path, update the routing table, and send BGP update packets. These packets force other routers in the topology to generate a series of routing

updates. Thus, a large number of BGP update packets are accompanied by frequent routing table changes, resulting in route flapping, which greatly degrades the network's performance. Therefore, it is necessary to build a large-scale network experimental platform to verify whether our traffic control method can emulate real-world scenarios.

We used the OpenStack cloud platform as the emulation environment to build a large-scale network topology and emulate an LDoS attack. A simple sketch of the topology is shown in Figure 8.

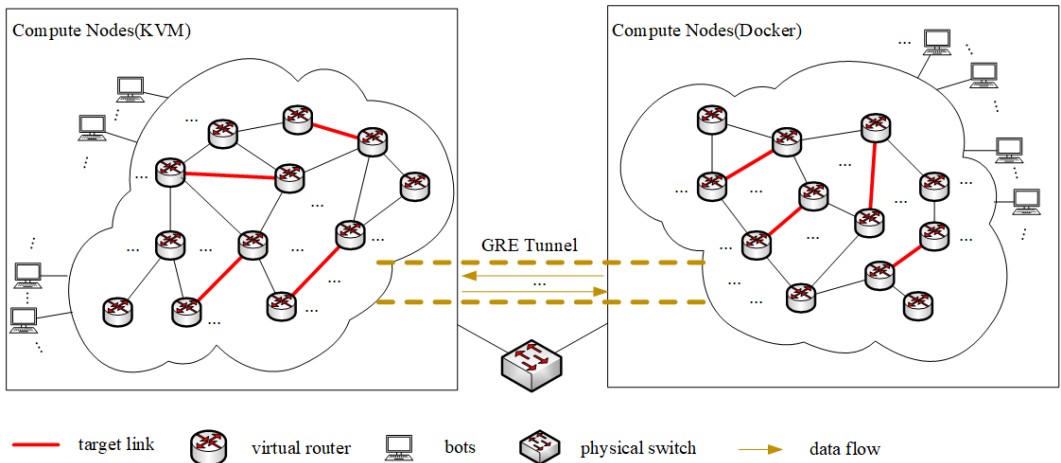

**Figure 8.** Topology of the large-scale LDoS experimental scenario.

Specifically, based on CAIDA AS Rank, we considered one AS domain as a virtual router, and divided the topology into 10 parts with METIS multi-layer graph segmentation technology. Then, on the OpenStack cloud platform which contains 10 computing nodes (which have installed Docker or KVM), we built a complex inter-AS network topology with 3000 virtual routers and 5179 links. Among the virtual routers, KVM was used to emulate only a small number of key routers because it consumes more resources but is more similar to a physical device; the other routers were emulated by Docker. In order to maximize the route flapping, we calculated the shortest paths between all the pairs of nodes in the topology by breadth first search and back tracking algorithm, and then got all the routing links. We selected 30 core links as the target links and simultaneously launched LDoS attacks on all of them. Any end of the target link can be KVM or Docker router, and the bandwidths of them are limited to 1 Gbps by our traffic control method. For each target link, the attack data stream was set as 3 Gbps. Each bot generates a periodic square-wave of UDP data stream with $R = 100$ Mbps, $L = 0.6$ s, and $T = 1$ s with the UDP traffic generator, and sends it to other bots through the target links.

In the one-hour attack emulation, 22 target links were interrupted; moreover, due to long-term congestion, some of the links were interrupted again after the BGP sessions had been reset. Overall, the LDoS attacks interrupted 135 target links. During the attack emulation, we also counted the number of BGP update packets and the number of times route flapping occurred in the entire topology. As shown in Figure 9, the attack caused 157,914 instances of route flapping throughout the entire topology, and a maximum of 454 routers experienced route flapping over a single second. Furthermore, a total of 17,528,093 update packets were generated with a single-second maximum of 47,333.

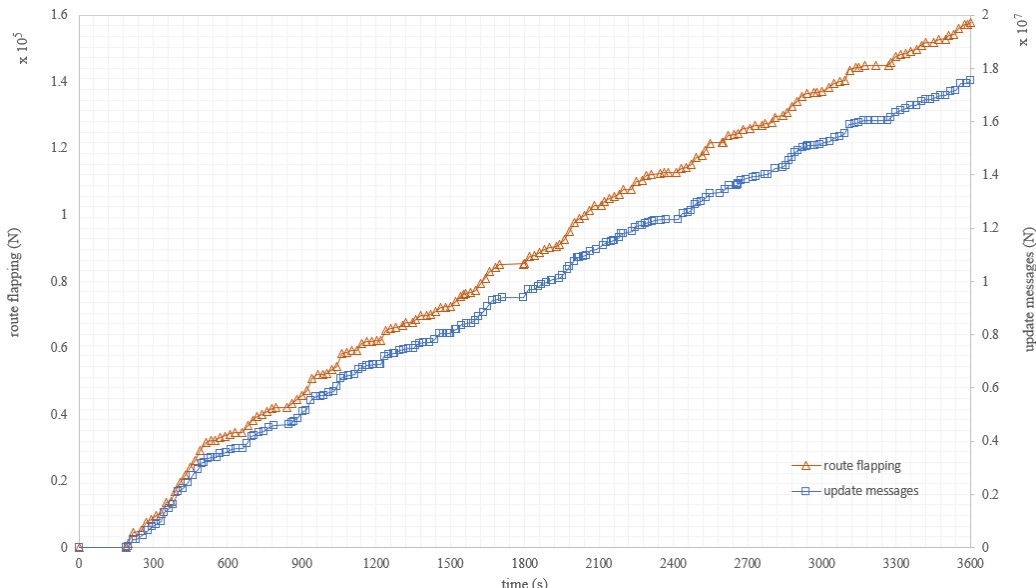

**Figure 9.** Results of the large-scale LDoS experimental scenario.

The experiments show that, on the one hand, when the network is relatively complex, an LDoS attack can force the routers in the topology to perform an immense number of route recalculations and transmit a very large number of BGP update packets, seriously affecting the network's performance. Compared with the experiments in the simple network topology, we fully considered the real situation of the complex network. Our virtual routers perform well under large-scale complex networks, and do not unpack the packet, so its security is good. Therefore, the virtual router based on our traffic control method can effectively support the research in congestion scenarios on the emulation platforms based on virtualization technology. On the other hand, because BGP-LDoS attack is launched from the data plane through large-scale traffic, the current defense strategy is mainly illegal traffic by detecting traffic characteristics. However, it is difficult to guarantee the reliability of the detection results and filter the illegal flow completely. Many proposed solutions are too idealistic to be verified in the emulation environment. But with the help of our high-fidelity emulation method, the research on the overall defense strategy of LDoS attack can be realized.

## 6. Conclusions and Future Work

This paper proposed a high-fidelity router emulation technology based on multi-scale virtualization. We first introduced the emulation architectures of the virtual routers, including the virtualization plane and routing plane. Accordingly, we emulated the traffic control method of a physical router using the drop from tail queue management method, the FIFO queue scheduling rule, and a bandwidth control method that effectively controls the packet transmission time. This strategy effectively overcomes the low fidelity of the conventional virtual router in a congestion scenario. At the same time, we separately designed two traffic control modules—one in the KVM kernel space and one in the Docker user space. Together, these achieve a high-fidelity and diversified route emulation solution.

Finally, we verified the effectiveness and fidelity of the proposed traffic control method by conducting several groups of experiments, including evaluations of the bandwidth control, the packet loss rate, and emulations of DDoS and LDoS attacks. Furthermore, through a large-scale LDoS emulation experiment, we verified the practicability of the KVM and Docker virtual routers constructed in this paper for use in the network security research domain.

In the future, we intend to construct a flexible and reliable network security experimental platform based on our router emulation technology that can be used for research on network attack and defense strategies and to evaluate their effects.

**Author Contributions:** Conceptualization, H.S. and X.W.; methodology, H.S. and M.Z.; software, H.S.; validation, H.S., M.Z., and G.Z.; writing—original draft preparation, H.S.; writing—review and editing, H.S.; project administration, X.W.; funding acquisition, X.W. All authors have read and agreed to the published version of the manuscript.

**Funding:** This research was funded by the National Natural Science Foundation of China (grant numbers 61672264 and 61972182), the National Key R&D Program of China (grant number 2016YFB0800801), and the Peng Cheng Laboratory Project of Guangdong Province (grant number PCL2018KP004).

**Conflicts of Interest:** The authors declare no conflict of interest.

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
