# Peer review of "High-Fidelity Router Emulation Technologies Based on Multi-Scale Virtualization†"

_information, doi:10.3390/info11010047_

Round 1

Reviewer 1 Report

This paper introduces a tool for emulating routers. Overall the authors presented the design and evaluation of their work nicely. However, I summarize the following issues for improvement. 

First, the introduction section needs to be revised. After reading the first two paragraphs, I did not see what problem(s) the authors are trying to solve. For example, "the virtual router constructed by virtualization, the routing software and the TC can satisfy most network emulation requirements" shows no more work needs to be done for router emulation? The next paragraph starting with "However, when emulating a large-scale denial of service" I think the authors were trying to convey that TC virtual routers have different behavior compared to a physical router, but this is not obvious to the reader. I suggest that the authors revise the first few paragraphs to make the points clear. 

In related work, the authors mentioned that serious distortion had a large impact on emulation results. I think this argument needs to be moved earlier in the paper to show why TC virtual routers have different behavior compared to a physical router. A few realistic examples (e.g., what kind of distortion) could improve the argument. 

The architecture in Figure 1 has two components for KVM and Docker. I think one is enough because the components look identical. In their design, the authors used equations (2), (3), (4), (5) as the regression formula. Would the numbers (e.g., 65.359) work for this particular case, or any delay time in general? The authors should explain how generic these formula are. 

In Section 5.4, the DDoS example has only 3 hosts on each side of the network. Since DDoS is distributed, 3 hosts would not be enough in such an attack. In Section 5.5, I don't think the statement "building high-fidelity virtual routers is critical for developing defense strategies against such attacks" is demonstrated by the work in this paper. Virtual routers introduced by the authors can produce a realistic environment, but not any defense strategies.

Reviewer 2 Report

Observations:

Even if the proposed method is functional, there is no security analysis to convince that actually the method can have practical usage. There is a possibility to present or depict an attack scenario and to make a short presentation of the method that can be used to resolve the attack?

In page 13-14 the LDoS and DDoS are presented. Can we have a detailed presentation on how the DDoS and LDoS attacks were conducted and which were the exactly results regarding the confidentiality, integrity and authenticity?

Figures

The reviewer wonder if figures are original or adapted from other works.
Figures 1–3, figures 5, 7 and 8.

Round 2

Reviewer 1 Report

The authors improved the paper by revising some sections and a larger experiment. The current paper is improved, except some writing issues (English grammar) in the first paragraph of the Introduction, and the last paragraph of Section 5.